# Comparative Analysis of Maternal Colostrum and Colostrum Replacer Effects on Immunity, Growth, and Health of Japanese Black Calves

**DOI:** 10.3390/ani14020346

**Published:** 2024-01-22

**Authors:** Marimu Urakawa, Mahmoud Baakhtari, Amany Ramah, Shoichiro Imatake, Parnian Ahmadi, Yuichiro Deguchi, Mizuho Uematsu, Yoshiki Nakama, Kazunari Imabeppu, Yusuke Nomura, Masahiro Yasuda

**Affiliations:** 1Laboratory of Veterinary Anatomy, Faculty of Agriculture, University of Miyazaki, Miyazaki 889-2192, Japanmahmoud.baakhtari@yahoo.com (M.B.);; 2Faculty of Veterinary Science, Balkh University, Mazar-i-Sharif 1703, Afghanistan; 3Department of Forensic Medicine and Toxicology, Faculty of Veterinary Medicine, Benha 13518, Egypt; 4Graduate School of Medicine and Veterinary Medicine, University of Miyazaki, Miyazaki 889-2192, Japan; 5Miyazaki Agricultural Mutual Aid Association, Miyazaki 880-0877, Japan

**Keywords:** colostrum, growth, health, immunity, Japanese black calf

## Abstract

**Simple Summary:**

The immunity, growth, and health of Japanese black calves was evaluated via feeding maternal colostrum versus colostrum replacer. Calves that received maternal colostrum showed significant increases in T cells and B cells in the peripheral blood. In addition, the expression levels of interleukin-1β–, interleukin-2–, and interferon-γ–encoding mRNAs were significantly higher in the maternal colostrum group. A lower incidence of disease and higher carcass weight in calves fed maternal colostrum were observed. These results suggest that, compared to maternal colostrum, the use of colostrum replacers may result in delayed immune system activation and slower growth in calves.

**Abstract:**

Maternal colostrum (MC) is an important source of nutrients and immune factors for newborn calves. However, when colostrum is unavailable or of poor quality, a colostrum replacer (CR) may be a suitable alternative to MC. As stock-raising farmers must make informed decisions about colostrum feeding management, this study was conducted to determine the effect of feeding MC versus CR on the promotion of immunological status, growth, and health in pre-weaned Japanese black (JB) calves. Sixteen newborn JB calves were fed MC after birth, and 16 JB calves were fed CR. For the MC group, the numbers of γδ T cells, CD4^+^ cells, CD8^+^ cells, CD4^+^CD8^+^ cells, B cells, and MHC class II^+^ cells were significantly higher compared with the CR group. Furthermore, the expression levels of interleukin (IL)-1β-, IL-2-, and interferon-γ (IFN-γ)-encoding mRNAs were significantly higher in the MC group compared with the CR group. A lower incidence of disease in 1-month-old calves and higher carcass weight in the MC group were observed compared with the CR group. These results suggest that CR activates the immune system delayed in calves compared with MC. MC increases populations of various immunocompetent cells, which can reduce infection rates and improve body weight gain.

## 1. Introduction

During the gestation period in cows, the transplacental transmission of immunoglobulins is impeded by the placental structure. Consequently, the acquisition of passive immunity in neonatal calves depends on maternal colostrum (MC) [1]. Calf health, future productive life, and farm profitability are directly related to the management of colostrum and the successful transfer of passive immunity [2]. Colostrum contains high concentrations of nutrients, immunoglobulins, and cytokines and is widely acknowledged for its role in bolstering disease resistance [3,4]. However, the quality and quantity of colostrum are contingent upon on the health status of the producing cow [5].

In instances in which colostrum is either unavailable or of poor quality, an appropriate alternative for MC would be a colostrum replacer (CR) [6]. The prevalent adoption of early weaning techniques has led to the widespread use of CR among livestock farmers. CR is designed to replace MC and encapsulates bovine immunoglobulins derived typically from colostrum or plasma [7]. However, the utilization of CR entails additional costs associated with milk replacement and requires supplementary efforts for the suckling of calves [8]. Although CR has advantages in terms of consistent composition and quality, its impact on the well-being of calves is characterized by variability [9,10].

The use of CR emerged from concerns regarding the potential presence of pathogenic agents in colostrum [7]. Previous studies documented a reduction in the incidence of Johne’s disease through the administration of CR [11]. Similarly, reports indicate a nearly 50% reduction in the chance of infection with *Mycobacterium avium* ssp. *paratuberculosis* through the feeding of CR [11]. However, reports conflict regarding the success of passive transfer when utilizing CR. Failure in the transfer of passive immunity is one of the most important risk factors and predictors of morbidity and mortality among dairy calves [12]. Poulsen et al. (2010) conducted a comparative analysis of passive transfer in bovines fed MC and CR and found no significant differences [13]. Conversely, Swan et al. (2007) reported the failure of passive transfer in calves fed CR due to the insufficient absorption of immunoglobulins, thereby potentially impairing health and survival [8]. To prevent or mitigate the failure of passive immunity transfer in neonatal Japanese black (JB) calves, information regarding the impact of MC and CR on immunological parameters is essential and would enable producers to make more-informed decisions regarding colostrum feeding management. Therefore, the purpose of this study was to determine the effect of feeding a commercially available CR versus MC on the immunological profile, growth, and health of pre-weaned JB calves.

## 2. Materials and Methods

All procedures used in this study were conducted according to protocols approved by the Animal Care and Use Committee of the University of Miyazaki (approval no. 2019-001-04).

### 2.1. Animals and Experimental Design

Newborn JB calves, sourced from a commercial JB dairy company in Miyazaki, Japan, which operates four farms (A, B, C, and D), were methodically allocated into two groups. Sixteen calves from farms A and B received MC. Simultaneously, 16 calves from farms C and D were administered CR. The dams were typically grouped in a large room of 3 to 4 cows and housed in separate rooms at the end of pregnancy and post-calving. Cows in maternity rooms were monitored continuously by veterinary staff to determine the calving time. Delivery rooms dedicated to pregnant cows were specifically designed on each farm. The enrolled calves were delivered naturally without any abnormalities and consistently monitored thereafter. Calves were allowed to stay in the maternity stall for the initial days. Calves were weaned at 60 days post-birth and then housed in groups of 5 to 6 animals in a designated room. All four farms adhered to a unified management system for JB calves from calving to market age. The history of respiratory diseases and treatment times during the first month of life were recorded in both groups by veterinary staff, and the carcass weight at 30 months of age was estimated after slaughtering at slaughterhouses.

### 2.2. Colostrum Feeding

MC was fed within the initial 10 h post-birth (maximum 3 L of colostrum) until weaning started. CR was fed within the initial 12 h post-birth (maximum 4.2 L containing 150 g of colostrum-derived IgG, which is a requisite amount of IgG for achieving successful passive transfer in newborn calves [14]) until weaning started. The CR (Elanco Co., Tokyo, Japan) was stored under refrigerated conditions before use and administered in three 750 g doses (between 2 h, between 6 h, and between 12 h) post-birth. Prior to administering CR, a container of bottled water was placed in a bucket filled with hot tap water for warming. Temperature was verified using a thermometer 10 min before feeding to ensure that the water reached the specified target of 41 °C. The mixing bucket, containing water, received 750 g of CR powder in three doses. Subsequently, the prepared mixture was transferred into feeder containers. Research technicians were responsible for the tasks associated with CR preparation. The CR was manufactured using carefully selected high-quality cow colostrum as a raw material. The inherent variability of colostrum is standardized to a certain quality to ensure a high level of safety, and the positive outcomes attributed to the utilization of the CR were evidenced in previous study [15].

### 2.3. Feeding Management

A uniform feeding regimen, consisting of starter grain, hay, and water, was introduced to calves 2 days post-birth, maintaining consistency throughout the perinatal period on all four farms. Dams and calves were treated using the standard feeding system for beef cattle [16]. During the fattening period, all calves were moved to a single large fattening area and fed a uniform amount of food. Throughout the pregnancy period, mother cows received 6 kg of hay and 1 kg of concentrated food, 113% total digestible nutrients (TDN) and 96% crude protein (CP). In the final 30 days before calving, the regimen was adjusted to 6 kg of hay and 2.5 kg of concentrated food (112% TDN and 94% CP). During the postpartum period following calving, dams were fed 6 kg of hay and 1 kg of concentrated food (TDN 141% and CP 133%) (Appendix A).

### 2.4. Sampling and Purification of Peripheral Blood Mononuclear Cells (PBMCs)

Blood samples were collected from newborn calves at the ages of 1, 2, and 3 months and transported from the farms to the University of Miyazaki in sterile tubes. Ficoll-Paque PLUS (GE Healthcare UK, Little Chalfont, Buckinghamshire, UK) and NH_4_Cl lysis buffer [17] were used to purify PBMCs and remove red blood cells, respectively. Finally, phosphate-buffered saline (PBS) (Fujifilm Wako Pure Chemical Co., Tokyo, Japan) was used to wash the remaining cells.

### 2.5. Lymphocyte Populations

PBMCs were suspended in PBS with 0.5% bovine serum albumin (Nacalai Tesque, Inc., Kyoto, Japan) and 0.05% sodium azide (Fujifilm Wako Pure Chemical Co., Tokyo, Japan) (BSA-PBS). The trypan blue exclusion test was used to detect and count viable cells (at densities ranging from 1 × 10^5^ to 1 × 10^6^ cells). Viable cells were then treated at 4 °C for 1 h with fluorescently labeled monoclonal antibodies (mAbs). After washing three times with BSA-PBS, the cells incubated with mAbs were resuspended in BSA-PBS. A FACS CantoTM II system (Becton Dickinson, Franklin Lakes, NJ, USA) was used to determine the relative immunofluorescence intensities of labeled cell populations. Anti-CD4 ILA11A, anti-CD8 CC63, anti-γδ TCR GB21A, anti-MHC class II TH14B (200-fold dilution, Monoclonal Antibody Center at Washington State University, Pullman, WA, USA), and anti-IgM BIG73A (100-fold dilution, Monoclonal Antibody Center at Washington State University) antibodies were used (Appendix A). A FITC labeling kit-NH2 (Dojindo Laboratories, Kumamoto, Japan), HiLyteTM Fluor 555 (F555) labeling kit-NH2 (Dojindo Laboratories), and HiLyteTM Fluor 647 (F647) labeling kit-NH2 (Dojindo Laboratories) were utilized to mark fluorescently labeled mAbs, as directed by the manufacturer. Positive cell density (cells/µL) was defined as the number of white blood cells (WBCs) counted using pocH-100iV Diff (Sysmex, Hyogo, Japan) × the percentage of lymphocytes counted by Giemsa staining of blood smears × the subset population determined by FACS.

### 2.6. Calculation of the Phagocytosis Index

At the age of 1 month, buffy coat was collected from blood samples. NH_4_Cl lysis solution was used to remove red blood cells. The remaining cells were rinsed with PBS, and RPMI 1640 medium (Fujifilm Wako Pure Chemical Co.) with 10% fetal calf serum (Sigma-Aldrich, St. Louis, MO, USA) and antibiotics (Fujifilm Wako Pure Chemical Co.) was used to suspend the cells (4.0 × 10^6^ cells/mL). The specific conditions (37 °C for 1 h in a 5% CO_2_ humidified atmosphere) were adjusted to incubate the mixture of cells with 2.5% suspension of latex beads suspension labeled with FITC (1 µm diameter, L1030, Sigma-Aldrich). Ice-cold 1 mM EDTA-PBS was used to remove cell-free beads from the mixture. The remaining cells were incubated with F647-labeled anti–MHC class II (TH14B) mAbs at 200-fold dilution, and F555-labeled anti-granulocyte (CH138A, Monoclonal Antibody Center at Washington State University) mAbs at 100-fold dilution (Appendix A), followed by analysis on a FACS CantoTM II system. The percentage of FITC^+^ TH14B^+^ cells relative to all TH14B^+^ cells and the percentage of FITC^+^ CH138A^+^ cells relative to all CH138A^+^ cells were used to determine the phagocytic index.

### 2.7. Proliferation of Lymphocytes

PBMCs (2 × 10^5^ cells/well) in RPMI 1640 medium from 1-month-old calves were stimulated at 37 °C for 72 h in a 5% CO_2_ humidified atmosphere with concanavalin A (Con A, Sigma-Aldrich) at a final concentration of 3.13 μg/mL and phytohemagglutinin (PHA; Sigma-Aldrich) at a final concentration of 15.6 μg/mL. A total of 20 μL of 3-(4,5-dimethylthiazol-2-yl)-2,5-diphenyl tetrazolium bromide (MTT, 5 mg/mL, Fujifilm Wako Pure Chemical Co.) was added to each well 4 h before the end of culture. At the end of culture, formazan crystals were collected and dissolved with dimethyl sulfoxide. A microplate reader (Benchmark Plus, measurement wavelength 570 nm, reference wavelength 610 nm, Bio-Rad, Hercules, CA, USA) was used to measure the optical density (OD), and the OD of the experimental group divided by the OD of the control group was defined as the stimulation index (SI) as a measure of lymphocyte proliferation.

### 2.8. RNA Extraction and Expression of mRNAs

An RNeasy Mini kit (Qiagen, Valencia, CA, USA) and one-step TB Green PrimeScript PLUS RT-PCR kit (Takara Bio., Tokyo, Japan) were used according to the manufacturers’ instructions to extract RNA from PBMCs from 1-month-old calves and perform real-time RT-PCR, respectively. Oligo7 software (Molecular Biology Insights, Colorado Springs, CO, USA) was used to design the primer pairs (Table 1). A QuantStudio™ Real-Time PCR system (Applied Biosystems, Carlsbad, CA, USA) was used, and the conditions for real-time PCR were as follows: reverse transcription for 5 min at 42 °C, initial PCR activation for 10 s at 95 °C, and 40 cycles of 5 s at 95 °C, 30 s at 57 °C, and 30 s at 70 °C; a dissociation curve was then generated according to the results. Glyceraldehyde phosphate dehydrogenase (GAPDH) was used to normalize the mRNA expression levels of other target genes, and the expression levels of GAPDH were not significantly different among the samples. The comparative Ct method (2^−ΔΔCt^ method/Livak method) and QuantStudio™ software v1.x series (Thermo Fisher Scientific, Waltham, MA, USA) were used to measure target mRNAs and analyze the data, respectively [18].

### 2.9. Statistical Analysis

Statistical analysis of differences between the MC and CR groups was performed using the Mann–Whitney *U*-test. Results are expressed as the mean ± SD. *p*-values < 0.05 were considered statistically significant. Statistical analysis was performed using the statistical software package SPSS for Windows (version 20.0, SPSS Inc., Chicago, IL, USA).

## 3. Results

### 3.1. Lymphocyte Subset Analysis

The numbers of CD4^+^ (Figure 1A), CD8^+^ (Figure 1B), and CD4^+^CD8^+^ (Figure 1C) cells in the MC group were significantly higher (*p* < 0.05) compared with the CR group at 1 month of age; however, there were no significant differences between the groups at 2 and 3 months of age. Additionally, the number of γδ T cells in the MC group was significantly higher (*p* < 0.05) than that in the CR group at 1 and 2 months of age (Figure 1D). Furthermore, the numbers of B cells (Figure 2A) and MHC class II^+^ cells (Figure 2B) in the MC group were significantly higher (*p* < 0.01 and *p* < 0.05, respectively) compared with the CR group at 3 months of age, but there were no significant differences at 1 and 2 months of age. The number of WBCs was slightly higher in the MC group for all 3 months compared to the CR group; however, the difference between groups was not significant (Appendix A).

### 3.2. Phagocytosis and Lymphocyte Proliferation

The phagocytic index of granulocytes and MHC class II^+^ monocytes did not differ significantly between the MC and CR groups (Figure 3A). Similarly, there was no significant change in the lymphocyte proliferative response to Con A and PHA between groups, but the responses to Con A and PHA were slightly higher in the MC group than the CR group (Figure 3B).

### 3.3. Expression of Cytokine-Encoding mRNAs

The expression of cytokine-encoding mRNAs in the MC and CR groups was analyzed at 1 month of age. The levels of IL-1β, IL-2, and IFN-γ mRNAs were significantly higher (*p* < 0.01, *p* < 0.01, and *p* < 0.05, respectively) in the MC group than the CR group (Figure 4).

### 3.4. Respiratory Disease Treatment and Carcass Weight Measurement

The frequency of treatment of calves for respiratory diseases decreased in the MC group. In 1-month-old calves, the frequency of treatments (Figure 5A) for respiratory diseases was significantly lower (*p* < 0.01) in the MC group than the CR group. Moreover, the carcass weight of calves in the MC group was significantly higher (*p* < 0.05) than that of calves in the CR group (Figure 5B).

## 4. Discussion

Colostrum immunity-related components such as lymphocytes are readily transferred to newborn calves, in which they circulate and stimulate neonatal immune functions [19,20,21]. In the present study, the numbers of CD4^+^, CD8^+^, and CD4^+^CD8^+^ cells in calves during the first month of life were significantly higher in the MC group. CD4^+^ cells are the primary cell fraction for acquired immunity, and memory CD8^+^ T cells are the principal component of immunity in the defense against viral infection [22]. Likewise, CD4^+^CD8^+^ cells reportedly play a role in several autoimmune diseases, virus infections, and cancer [23,24,25]. Regarding the pro-inflammatory and anti-inflammatory functions of CD4^+^CD8^+^ cells, Diedrich et al. [26] suggested in response to *Mycobacterium tuberculosis* infection that CD4^+^CD8^+^ cells play an important role in immune reactions. The observed increases in lymphocyte subset populations in newborn calves suggest that components present in MC enhance calf immune responses.

A high proportion of γδ T cells among the circulating lymphocyte population is a characteristic of healthy immune conditions in young ruminants [27]. In our study, the number of γδ T cells in the MC group was significantly higher in 1- and 2-month-old calves. Lundberg et al. [28] and Murakami et al. [29] reported that γδ T cells in peripheral blood play an important role in anti-viral responses. γδ T cells also appear to play an important role in bridging the innate and adaptive immune responses [30]. The low number of γδ T cells observed in the CR group in the present study might be unfavorable for the immune system. Fewer γδ T cells may be associated with susceptibility to infection in JB calves [31]. Therefore, the increase in the number of γδ T cells resulting from MC consumption may enhance immunity in calves after birth and increase their disease resistance.

Blood mononuclear leukocyte populations in newborn calves are characterized by a higher proportion of γδ T cells [32,33] and lower proportion of B cells [34]. In our study, the numbers of B cells and MHC class II^+^ cells in MC group calves were lower at 1 and 2 months of age compared with 3 months. At the age of 3 months, the numbers of these cells were significantly higher compared with the CR group. The percentage of B cells increased over time, most notably between 32 and 60 days of age. Similar age-dependent increases in the proportion of circulating B cells in young calves have been reported [34], which is typical of the maturing immune system in calves. Thus, the increases in B cell and MHC class II^+^ cell populations in 3-month-old calves suggest that MC contributes to the development of calf immunity even after passive immunity has elapsed.

Phagocytosis plays an important role in early host defense, and some bioactive factors in colostrum, such as immunoglobulins [35], complement components [36], lactoferrin [37], cytokines [33], and other as yet unknown factors [38], are thought to function as phagocytosis-activating agents. In this study, the phagocytic indexes of granulocytes and monocytes did not differ significantly between the MC and CR groups. It was previously reported that human colostrum contains a phagocytosis-promoting factor [39], and the absence of any significant changes in phagocytosis between the MC and CR groups could be related to the short lifespan of neutrophils. Neutrophils, with a shorter lifespan compared to lymphocytes [40], may exert less of an effect in colostrum on phagocytic efficacy by the time samples are collected at 1 month of age. In addition, in response to stimulation of lymphocytes by Con A and PHA, no significant changes were observed between the MC and CR groups. A previous study reported that increasing the nutrition regime to a higher level in young calves has no effect on improving the general responsiveness of PBMC populations in response to mitogenic stimulation [41]. Therefore, the present results suggest that both MC and CR may provide sufficient protein and energy to maintain or promote the general responsiveness of lymphocyte proliferation.

Immunoglobulins are one of the main components of total protein in blood and play a pivotal role in fostering both passive and adaptive immune responses in neonatal animals. MC is an important source of diverse immunoglobulins and facilitates their transfer to newborn animals through feeding. Therefore, to evaluate the efficacy of colostrum, it is essential to measure gene expression levels and the protein concentration. Recent studies in calves and piglets have demonstrated a significant increase in protein concentrations, particularly the concentration of immunoglobulins, following the administration of MC to neonatal animals [42,43]. In the context of immune enhancement by colostrum, we extended the investigation to the expression levels of cytokine-encoding mRNAs in the present study. MC significantly stimulated the systemic expression of transcripts encoding IFN-γ, IL-1β, and IL-2 compared with CR. IFN-γ and IL-1β are secreted by Th1 cells [44] and innate immune cells such as macrophages [45], respectively. IL-1β contributes to the immune response against pathogens by inducing fever, activating lymphocytes, and promoting the infiltration of leukocytes into sites of infection [46]. Likewise, IFN-γ is recognized as an important factor for preventing bacterial invasion of mucous membranes [47]. Moreover, IL-2 is necessary for the development of T cell memory and the development and maturation of regulatory T cells in the thymus [48,49]. Therefore, it is possible that calves in the CR group with a lower exposure to IL-2 during the first month of life might exhibit the reduced development of T cell memory and differentiation. During the first month of life, in which the development of Th1 cells and regulation of innate immune cell responses occur, MC plays a fundamental role that may assist in reducing the risk of infectious illness.

Quigley et al. (2001) and Jones et al. (2004) observed higher feed efficiency in calves fed MC. A tendency for MC-fed calves to gain more BW in the first week of life has also been reported [9,50]. In our study, calves fed MC had significantly greater carcass weight than calves fed CR. Therefore, the present study confirms the growth-promoting benefits of feeding MC to JB calves. In addition, when disease was present, the CR group exhibited a higher number of treatment days and frequency of treatments compared with the MC group at 1 month of age. It was previously reported that 93% of CR-fed calves exhibited a failure of passive transfer, and the negative effect of failure of passive transfer on morbidity and mortality is well known in neonatal calves [51,52]. Similarly, another study reported significantly lower morbidity and mortality in calves fed MC (46.9%) compared with calves fed plasma-derived (71.4%) or colostrum-derived (67.3%) CR [53]. The greatest risk of calf morbidity and mortality is observed during the first month of life [54,55,56]; therefore, one of our study objectives focused on reducing mortality and morbidity in 1-month-old JB calves. It is thus important to pay attention to the frequency of treatments in the first month of a study. In the present study, local and systemic protection provided by immunoglobulins and immunomodulatory molecules contained in MC might have reduced the incidence of infection and therefore the need for antibiotic therapy in calves fed MC. The decreased incidence rates of respiratory diseases in MC-fed JB calves suggest that MC increases disease resistance by enhancing both natural and acquired immunity in the first month after birth.

## 5. Conclusions

JB calves fed CR exhibited the delayed activation of the immune system and higher chances of failure in the transfer of passive immunity. In addition, the feeding of CR promoted decreases in populations of various immunocompetent cells not only during the period of colostrum feeding but also after passive immunity disappeared. Feeding JB calves CR may increase the chances of bacterial infection in early life while simultaneously diminished calf performance post-colostrum, as evidenced by reduced disease resistance, increased infection rates, and decreased carcass weight.

## Figures and Tables

**Figure 1 animals-14-00346-f001:**
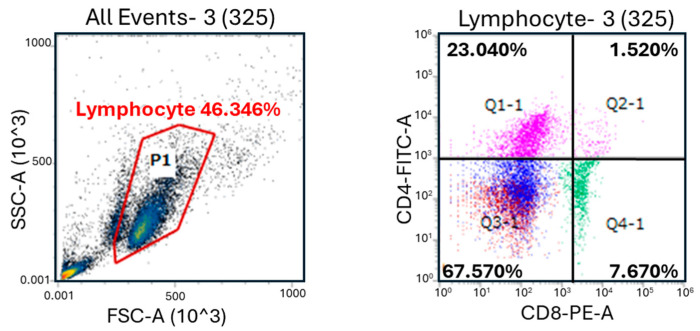
Relative population of CD4^+^ and CD8^+^ cells in the blood of newborn JB calves. The lymphocyte gate (P1) using forward scatter (FSC) and side scatter (SSC) for flow cytometry is shown in the upper left. Two-parameter dot pots used for gating the single- (Q1-1 and Q4-1) and double-positive (Q2-1) cell populations are shown in the upper right. The effect of MC and CR feeding on T cell populations in JB calves and populations of CD4^+^ cells (**A**), CD8^+^ cells (**B**), CD4^+^CD8^+^ cells (**C**), and γδ T cells (**D**) were recorded in 1-, 2-, and 3-month-old calves. Data are presented as the mean ± SD. Letters (a: *p* < 0.05 MC vs. CR) indicate significant differences.

**Figure 2 animals-14-00346-f002:**
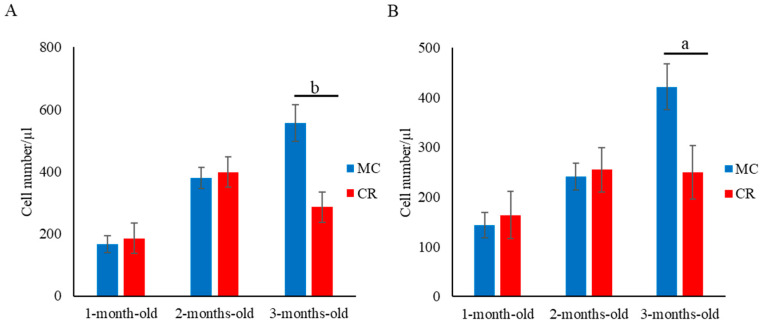
Effect of MC and CR feeding on B cell populations in JB calves. Populations of B cells (**A**) and MHC class II^+^ cells (**B**) were recorded in 1-, 2-, and 3-month-old calves. Data are presented as the mean ± SD. Letters (a: *p* < 0.05 and b: *p* < 0.01 MC vs. CR) indicate significant differences.

**Figure 3 animals-14-00346-f003:**
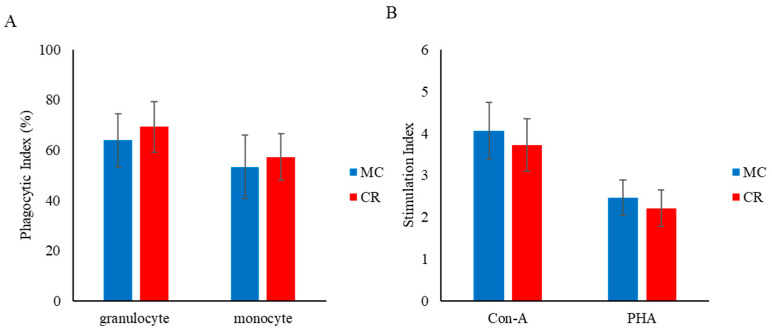
Effect of MC and CR feeding on the phagocytic index of granulocytes and monocytes (**A**) and effect of MC and CR feeding on lymphocyte proliferation in response to Con A and PHA treatment in JB calves (**B**). The phagocytic and stimulation indexes of mitogens were recorded in 1-month-old calves. Data are presented as the mean ± SD.

**Figure 4 animals-14-00346-f004:**
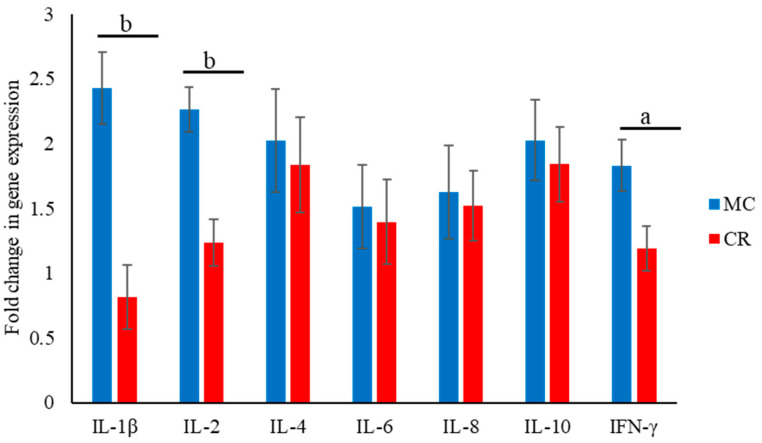
Effect of MC and CR feeding on the expression of mRNAs encoding IL-1β, IL-2, IL-4, IL-6, IL-10, and IFN-γ. The fold-change in the gene expression was measured in 1-month-old JB calves. Expression levels were measured using real-time PCR. Data are presented as the mean ± SD. Letters (a: *p* < 0.05 and b: *p* < 0.01 MC vs. CR) indicate significant differences.

**Figure 5 animals-14-00346-f005:**
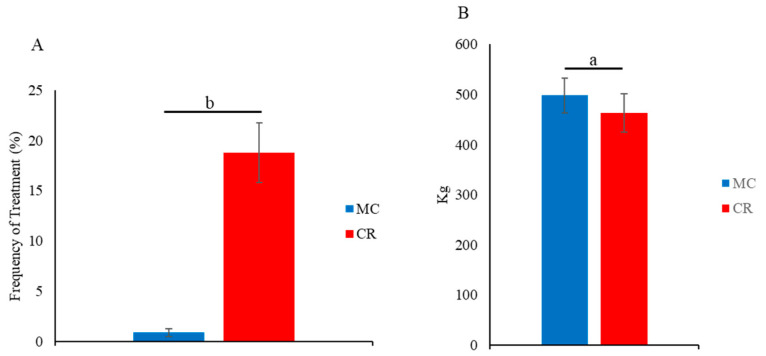
Effect of MC and CR feeding on treatment for respiratory diseases and carcass weight in JB calves. The frequency of treatments was recorded in 1-month-old JB calves by the Japanese Mutual Aid Association (NOSAI) veterinary staff (**A**), and carcass weight was recorded at the age of 30 months after slaughtering (**B**). Data are presented as the mean ± SD. Letters (a: *p* < 0.05 and b: *p* < 0.01 MC vs. CR) indicate significant differences.

**Table 1 animals-14-00346-t001:** Sequences of primers used for real-time PCR.

Gene	Primer	Sequences	Accession Number	Length
				(Base Pairs)
GAPDH	F	GTTCAACGGCACAGTCAAGGCAGAG	NM_001034034	123
	R	ACCACATACTCAGCACCAGCATCAC		
IL-1 β	F	GCCTACGCACATGTCTTCCA	NM_174093	111
	R	TGCGTCACACAGAAACTCGTC		
IL-2	F	TGCTGGATTTACAGTTGCTT	XM_024976996	111
	R	TCAATTCTGTAGCGTTAACCT		
IL-4	F	ATCAAAACGCTGAACATCCTC	NM_173921	142
	R	TCCTGTAGATACGCCTAAGCTC		
IL-6	F	AGCTCTCATTAAGCGCATGG	NM_173923	168
	R	ATCGCCTGATTGAACCCAG		
IL-10	F	GGCCTGACATCAAGGAGCAC	NM_174088	103
	R	CTCTTGTTTTCGCAGGGCAGA		
IFN-γ	F	TGATTCAAATTCCGGTGGAT	NM_174086	108
	R	TCTTCCGCTTTCTGAGGTT		

## Data Availability

The data presented in this study are available on request from the corresponding author.

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
