# Peer review of "Comparative Analysis of Maternal Colostrum and Colostrum Replacer Effects on Immunity, Growth, and Health of Japanese Black Calves"

_animals, 2024, doi:10.3390/ani14020346_

Round 1

Reviewer 1 Report (Previous Reviewer 1)

Comments and Suggestions for Authors

The author made the required changes to allow for data comprehension and replication. As a result, the article is ready for publishing.

Author Response

We appreciate Reviewer the time and effort that you dedicated to providing feedback on our manuscript, as well as your insightful comments and valuable improvements to our manuscript. 

Reviewer 2 Report (Previous Reviewer 3)

Comments and Suggestions for Authors

Appropriate additions were made in response to the points raised.

There is an error in the added line in the response letter, but it does not affect the main text.

We believe this is a research report worth publishing, as it contains a lot of useful information on farms about how to feed colostrum to calves.

Author Response

We appreciate Reviewer the time and effort that you dedicated to providing feedback on our manuscript, as well as your insightful comments and valuable improvements to our manuscript. 

Reviewer 3 Report (New Reviewer)

Comments and Suggestions for Authors

The manuscript addresses an important topic in cattle production, which is the use of colostrum replacers, with widespread use throughout the world, particularly in countries with more developed production systems, namely dairy.

The research design is appropriate, the methods are adequately described, and the results are clearly presented, emphasizing the limitations of colostrum replacers, information that all stakeholders in the sector must keep in mind, with correct management of maternal colostrum continuing to be the best way to achieve adequate rearing of cattle.

However, I have some concerns, namely about the way the authors approach this topic.

Maybe it's a current trend, but I find it strange that "promotion" is written when referring to the logical use of a natural product perfected over hundreds of millions of years, such as maternal colostrum.

Although it may be necessary to use colostrum replacers due e.g. to unavailability or poor quality of the maternal colostrum, the latter should still be considered the gold standard. Thus, the approach should focus on the limitations of the former compared to maternal colostrum regarding immunity, growth, and health.

Therefore, the title seems somewhat redundant. Some suggestions for titles that would perhaps make more sense:

- "Maternal colostrum or colostrum replacer? Assessing their impact in immunity, growth, and health in Japanese black calves";

- "Impact of maternal colostrum vs. colostrum replacer on immunity, growth, and health of Japanese black calves";

- "Comparative analysis of maternal colostrum and colostrum replacer effects on immunity, growth, and health of Japanese black calves";

"Colostrum choices: evaluation/assessment of their Influence on immune function, growth, and health in Japanese black calves".

In line 18, replace “The promotion of immunity” with “The immunity”.

The manuscript should perhaps address the differences more with the limitations of colostrum replacer than with the advantages of maternal colostrum. Therefore, in lines 23-25, I suggest replacing “These results suggest that, compared with colostrum replacer, maternal colostrum activates the immune system earlier and promotes growth in calves” with e.g. “These results suggest that, compared to maternal colostrum, the use of colostrum replacers may result in delayed immune system activation and slower growth in calves”.

Although the manufacturer of the colostrum replacer is referred (Elanco Co., Tokyo, Japan), I did not find any reference to the composition of the colostrum replacer, which for me is a very important gap in this manuscript. The Discussion itself should resort to differences in composition as a cause for differences in results, and the only one that is addressed is the presence of lymphocytes in maternal colostrum, obviously absent in the colostrum replacer.

Even though “all four farms adhered to a unified management system for JB calves from calving to market age”, I have some concerns regarding the separation by farm of the groups of calves fed maternal colostrum (A and B) and colostrum replacer (C and D). In the case of respiratory diseases, it is current knowledge that they are multifactorial, with the environment playing a substantial role in their manifestation.

The Conclusions should perhaps focus more on the limitations of colostrum replacers than on the advantages of maternal colostrum.

Most bibliographical references are not very recent, and I noticed the absence of some that I consider almost unavoidable on this topic, such as, e.g., Godden et al (2019). Colostrum management for dairy calves. Veterinary Clinics: Food Animal Practice, 35(3), 535-556.

Author Response

We appreciate Reviewer the time and effort that you dedicated to providing feedback on our manuscript, as well as your insightful comments and valuable improvements to our manuscript. All changes are now highlighted in red letters in the revised version of manuscript. Following are our responses to the Reviewer’s comments.

Reviewer #3

  • Change title.

Response: Thank you for your valuable suggestion regarding the title. Your input is greatly appreciated, and we believe that the suggested title accurately reflects the essence of our research. The chosen title, "Comparative analysis of maternal colostrum and colostrum replacer effects on immunity, growth, and health of Japanese black calves" aligns with the content of the manuscript. It clearly communicates the focus of the study, comparing the effects of maternal colostrum and colostrum replacer on various aspects of Japanese black calves, including immunity, growth, and health. It seems like a suitable and informative title for the research.

  • In line 18, replace “The promotion of immunity” with “The immunity”.

Response: “The promotion of immunity” replaced with “the immunity” in line 19.

  • The manuscript should perhaps address the differences more with the limitations of colostrum replacer than with the advantages of maternal colostrum. Therefore, in lines 23-25, I suggest replacing “These results suggest that, compared with colostrum replacer, maternal colostrum activates the immune system earlier and promotes growth in calves” with e.g. “These results suggest that, compared to maternal colostrum, the use of colostrum replacers may result in delayed immune system activation and slower growth in calves”.

Response: Thank you for your suggestion, we replaced the sentences in line 24-26 and 38-39 considering your suggestion regarding with the limitations of colostrum replacer than with the advantages of maternal colostrum.

  • Although the manufacturer of the colostrum replacer is referred (Elanco Co., Tokyo, Japan), I did not find any reference to the composition of the colostrum replacer, which for me is a very important gap in this manuscript. The Discussion itself should resort to differences in composition as a cause for differences in results, and the only one that is addressed is the presence of lymphocytes in maternal colostrum, obviously absent in the colostrum replacer.

Response: Thank you for your insightful feedback. We appreciate your keen observation regarding the absence of specific details on the composition of the colostrum replacer in our manuscript. Regrettably, we encountered limitations in obtaining specific details about the exact composition of the colostrum replacer from the manufacturer, Elanco Co., Tokyo, Japan. However, we would like to provide additional information based on the statement provided by the manufacturer, Elanco Co., Tokyo, Japan. “The colostrum replacer is manufactured using carefully selected high-quality cow colostrum as a raw material. The inherent variability of colostrum is standardized to a certain quality to ensure a high level of safety. The manufacturing process retains immunoglobulins, milk fat, and various nutritional components present in colostrum. The resulting powder is easily soluble, and the product boasts a long-term shelf life of three years from the date of manufacture.” As was mentioned in product detail the CR which was used in our study was made from high quality cow colostrum.

 In addition to this information, we would like to highlight that the effects of this colostrum replacer were addressed in another referenced study such as (Reference 15). While we acknowledge that specific quantitative details and potential variations are not explicitly addressed, the reference study provides insights into the practical application and outcomes of using this colostrum replacer. We believe that referencing this study will contribute to a better understanding of the composition and effects of the CR. We will incorporate both sets of information into our manuscript to provide a more comprehensive overview of the colostrum replacer, considering both its composition and its practical implications based on relevant research. Thank you for guiding us toward enhancing the clarity and completeness of our study. New information and citation were added in line 112-115.

  • Even though “all four farms adhered to a unified management system for JB calves from calving to market age”, I have some concerns regarding the separation by farm of the groups of calves fed maternal colostrum (A and B) and colostrum replacer (C and D). In the case of respiratory diseases, it is current knowledge that they are multifactorial, with the environment playing a substantial role in their manifestation.

Response: We appreciate the reviewer's attention to the grouping of calves from different farms in our study. We would like to clarify that all four farms under consideration are closely affiliated, belonging to the same company, and are situated near to each other in Miyazaki, Japan. The responsible staff and veterinarian diligently and consistently monitored both the management practices and environmental conditions throughout the study period. Importantly, there are no discernible differences in the environmental conditions across these farms. Additionally, these farms strictly adhere to a unified management system specifically tailored for Japanese black calves in the Miyazaki region. This comprehensive system ensures a consistent and standardized approach to calf rearing practices, minimizing potential variations in environmental factors that could influence the study outcomes. We believe that the uniformity in environmental conditions and management practices across the farms strengthens the internal validity of our study, allowing for a more robust assessment of the impact of maternal colostrum versus colostrum replacer on immunity, growth, and health in Japanese black calves.

  • The Conclusions should perhaps focus more on the limitations of colostrum replacers than on the advantages of maternal colostrum.

Response: Thank you for your valuable feedback. We have revisited the conclusions, placing a greater emphasis on highlighting the limitations of colostrum replacers rather than primarily emphasizing the advantages of maternal colostrum. We appreciate your suggestion and believe this adjustment enhances the overall balance and thoroughness of the manuscript. The changes were applied in line 414-420.

  • Most bibliographical references are not very recent, and I noticed the absence of some that I consider almost unavoidable on this topic, such as, e.g., Godden et al (2019). Colostrum management for dairy calves. Veterinary Clinics: Food Animal Practice, 35(3), 535-556.

Response: Thank you for your insightful observation. We acknowledge the importance of recent references in the field, including the work of Godden et al. (2019) on colostrum management for dairy calves. We appreciate your suggestion and have incorporated more recent references, including the mentioned publication, to ensure the inclusion of the latest and most relevant information in the revised manuscript. The cited article was mentioned in line 52-53 reference 6.

In addition, we omitted some of the old bibliographical references such as:

  • Kruse, P.E. The Importance of Colostral Immunoglobulins and Their Absorption from the Intestine of the Newborn Animals. Ann. Rech. Vet. 1983, 14, 349–353.
  • Robison, J.D.; Stott, G.H.; DeNise, S.K. Effects of Passive Immunity on Growth and Survival in the Dairy Heifer. J. Dairy Sci. 1988, 71, 1283–1287, doi:10.3168/jds.S0022-0302(88)79684-8.
  • Bush, L.J.; Staley, T.E. Absorption of Colostral Immunoglobulins in Newborn Calves. J. Dairy Sci. 1980, 63, 672–680, doi:10.3168/jds.S0022-0302(80)82989-4.
  • Senogles, D.R.; Muscoplat, C.C.; Paul, P.S.; Johnson, D.W. Ontogeny of Circulating B Lymphocytes in Neonatal Calves. Res. Vet. Sci. 1978, 25, 34–36.
  • Sasaki, M.; Davis, C.L.; Larson, B.L. Production and Turnover of IgG1 and IgG2 Immunoglobulins in the Bovine around Parturition. J. Dairy Sci. 1976, 59, 2046–2055, doi:10.3168/jds.S0022-0302(76)84486-4.
  • Larson, B.L.; Heary, H.L.; Devery, J.E. Immunoglobulin Production and Transport by the Mammary Gland. J. Dairy Sci. 1980, 63, 665–671, doi:10.3168/jds.S0022-0302(80)82988-2.

This manuscript is a resubmission of an earlier submission. The following is a list of the peer review reports and author responses from that submission.

Round 1

Reviewer 1 Report

Comments and Suggestions for Authors

Summary- The manuscript “Promotion of immunity, growth, and health of Japanese black 2 calves by feeding maternal colostrum” submitted by Urakawa, Yasuda, and colleagues has observed the influence of colostrum intake during the first 24 hours on calves' immunity. Overall, the manuscript is well written.

Comments and Suggestions are mentioned below –

Major – 1. One of the major concerns of the study is that some of the results reported by the author have already been published before in two of the studies (PMID: 36115917 and PMID: PMID: 36851432). Surprisingly, the author has not cited these articles.

2. It is very difficult to interpret cells data in graphs. Please, indicate the specific location on the graph where the T-cell of interest is being identified, along with the corresponding cluster that has been subjected to analysis. There is little point in discussing T cells without specifying their surface markers, such as CD3+, CD3+CD4+, and so on.

3- Oral tolerance is an early life event and disruption in this could result in immunopathology so, it would be interesting to see if the presence/absence of colostrum during the first 24h results in loss of oral tolerance to the dietary antigens later. The author should look at the levels of IFN-g in the serum of the calves.

4- Looking at the cytokine profiles in Fig 4, it looks like in MC group one of the calves have higher cytokine response for some reason as evident arguably from the outlier in IL-1beta, IL-2, IL-4, IL-6, IL-8, IL-10 and IFN-g. Can this be excluded as it could be skewing the statistical analysis? What does author think? Can they trace the single animal and check if it had any immunological or health abnormality by any chance such as loss in body weight and diarrheic episodes?

5- The information in the M&M on the panels and the used antibodies would best be given in a table, detailing marker, clone, IgG subclass, dilution and fluorochrome used to detect that marker. In addition, the authors should provide a figure showing the gating strategy for all used panels.

6- Please, provide a scatter plot in all figures.

7-The degree of correlation between gene expression and protein levels can vary among different cytokines/chemokines. It can be explained by posttranscriptional and posttranslational regulation and by misclassification due to measurement errors. Why did the authors not perform protein measurement of cytokines? In the course of the discussion, it is important to acknowledge the potential divergence in the results pertaining to gene and protein expression of cytokines, as compared to previous research endeavors.

Minor revisions:

1- Line 84: The author did not explain why 4.2 L contains 150 g of colostrum-derived IgG.

2- It is unclear to me how the authors can exclude doublets and dead cells based (propidium iodide).

3- on the CD3+CD4+CD8+ T cells is in contrast to the results shown in figure 1. This seems strange as this T cell subset is associated with memory T cells and antigen encounter. One would expect that this subset is higher in MC as compared to CR.

4- is too strong as effector cell functions were not investigated.

5- what was the antibody and markers used for B lymphocytes?

6- I noticed most of the antibodies are directly conjugated, but did the authors include isotype control

7-the caption for figure 1 appeared above the figure

8- how much of formula replacer was given to each calves? How many times?

9- Is it possible that these CR animals received a much more volume of colostrum replacer considering the capacity of milk production of cows?

10- could you also add the brand and origin of the tubes? Also, were the tubes of colostrum replacer kept refrigerated or at room temperature?

11- Were all of the cows multiparous?

Comments on the Quality of English Language

No comments

Author Response

We appreciate Reviewers the time and effort that you dedicated to providing feedback on our manuscript and are grateful for the insightful comments on and valuable improvements to our paper. All the changes are now highlighted in red letters in the revised version of our manuscript. The following are our responses to the Reviewer’s comments.

Reviewer1#

Major comments

  1. One of the major concerns of the study is that some of the results reported by the author have already been published before in two of the studies (PMID: 36115917 and PMID: PMID: 36851432). Surprisingly, the author has not cited these articles.

Response: "Thank you for bringing attention to these studies (PMID: 36115917 and PMID: 36851432). We would like to clarify that our manuscript was prepared prior to the publication of these articles. Unfortunately, due to unforeseen delays in the submission process, our paper could not be submitted to the journal in a timely manner. We acknowledge the importance of these referenced studies and their relevance to our work. In the revised version, we will include proper citations to acknowledge and integrate these relevant findings into our discussion. We appreciate your diligence in reviewing our manuscript and bringing this matter to our attention. We used the information about protein concentration in our manuscript and cited them in line 339-349.

  1. It is very difficult to interpret cells data in graphs. Please, indicate the specific location on the graph where the T-cell of interest is being identified, along with the corresponding cluster that has been subjected to analysis. There is little point in discussing T cells without specifying their surface markers, such as CD3+, CD3+CD4+, and so on.

Response: We mention the location of T cell of interest which was being identified along with the corresponding cluster that has been subjected to analysis in (Figure1), and regarding the information about antibodies it was mentioned in (Table S2).

3- Oral tolerance is an early life event and disruption in this could result in immunopathology so, it would be interesting to see if the presence/absence of colostrum during the first 24h results in loss of oral tolerance to the dietary antigens later. The author should look at the levels of IFN-g in the serum of the calves.

Response: Thank you for your thoughtful suggestion regarding the potential impact of colostrum on oral tolerance and the levels of IFN-g in the serum of the calves. Oral tolerance which is a state of systemic unresponsiveness that is the default response to food antigens in the gastrointestinal tract, while we agree that investigating the relationship between colostrum and oral tolerance is an intriguing avenue for future research, our current manuscript is specifically concentrated on examining the effect of colostrum on the promotion of immune status in Japanese black calves. As such, delving into the details of oral tolerance and IFN-g levels is beyond the scope of our present work. We appreciate your valuable input and will certainly consider this as a promising direction for separate research in the future. If you have any further recommendations or insights related to our current focus, we would be grateful for your guidance.

4- Looking at the cytokine profiles in Fig 4, it looks like in MC group one of the calves have higher cytokine response for some reason as evident arguably from the outlier in IL-1beta, IL-2, IL-4, IL-6, IL-8, IL-10 and IFN-g. Can this be excluded as it could be skewing the statistical analysis? What does the author think? Can they trace the single animal and check if it had any immunological or health abnormality by any chance such as loss in body weight and diarrheic episodes?

Response: The animals were checked daily by professional staff and veterinarians in all four farms and the responsible veterinarians did not record any severe health abnormalities or loss in body weight and diarrhea in none of the animals. The disease which veterinarian recorded in the history of calves during our study did not statistically affect the result. Also, by removing or omitting the data for single animal with higher cytokine response still we will get the same result for evaluating the level of mentioned cytokines.

5- The information in the M&M on the panels and the used antibodies would best be given in a table, detailing marker, clone, IgG subclass, dilution and fluorochrome used to detect that marker. In addition, the authors should provide a figure showing the gating strategy for all used panels.

Response: The table with antibodies information was added to our manuscript in (Table S2). Regarding the gating strategy, the gating is equal for measuring the lymphocyte population in all experiments, therefore we added one of them in (Figure 1).

6- Please, provide a scatter plot in all figures.

Response: With respect to your feedback, sorry I am not sure that we correctly understand your comment, However, after careful deliberation, we believe that conveying the information through bar charts might enhance the clarity and ease of understanding for our audience. Bar charts can offer a more straightforward visualization, especially when comparing multiple data points. We would be happy to discuss this further and are open to your insights on the matter. Please let us know if you have any specific concerns or if there were aspects you would like us to reconsider. With respect to your opinion, I think if you mean to use scatter plot instead of figures it would be bit difficult to understand the study.

7-The degree of correlation between gene expression and protein levels can vary among different cytokines/chemokines. It can be explained by posttranscriptional and posttranslational regulation and by misclassification due to measurement errors. Why did the authors not perform protein measurement of cytokines? In the course of the discussion, it is important to acknowledge the potential divergence in the results pertaining to gene and protein expression of cytokines, as compared to previous research endeavors.

Response: Thank you for your comment, in our study we tried to focus on promotion of growth, health and immunity by feeding colostrum in Japanese black calves. We evaluated the immune status of calves by specifying the subset population of critical immune cells and cytokine-encoding mRNAs level. Evaluating the immune status and mRNAs levels in calves could help us to conclude successful passive immunity transfer from mother cow to calves by colostrum. Therefore, we did not measure the protein concentration in this study. However, some previous publications measure the level of different protein concentration in newborn animals including calves, and we used those data in our discussion in line 339-349.

Minor revisions

1-Line 84: The author did not explain why 4.2 L contains 150 g of colostrum-derived IgG. 

Response: It is recommended that a calf needs to ingest at least 150 to 200 g of IgG within the first 12 hours of their lives to achieve successful passive transfer and it can be achieved by feeding 3 to 4 L of good quality colostrum. The information with reference was added in line 85-88.

2- It is unclear to me how the authors can exclude doublets and dead cells based (propidium iodide).

Response: Regrettably, propidium iodide was inadvertently added to the experimental procedure. Upon recognizing this error, we have promptly removed any reference to propidium iodide from the main text. We apologize for any confusion this may have caused and assure you that the correction has been made. We did not use propidium iodide for blood samples we usually use it in tissue samples to exclude dead cells. For blood samples we just added BSA-PBS without propidium iodide. The term of propidium iodide was deleted.

3- on the CD3+CD4+CD8+ T cells is in contrast to the results shown in figure 1. This seems strange as this T cell subset is associated with memory T cells and antigen encounter. One would expect that this subset is higher in MC as compared to CR.

Response: We appreciate the reviewer's attention to detail, particularly in examining the representation of CD4+CD8+ double positive T cells in our study involving calves. While we acknowledge the expectation that this T cell subset, associated with memory T cells and antigen encounters, might exhibit a higher representation in calves fed maternal colostrum (MC) compared to colostrum replacer (CR), our comprehensive analysis considered various factors that could influence these results.

CD4+CD8+ double positive T cells are more commonly associated with the thymus during T cell development, their presence in the peripheral blood, including in calves, is not as common. In the thymus, these cells undergo positive and negative selection before differentiating into either CD4+ or CD8+ single-positive T cells. However, if CD4+CD8+ T cells are found in the peripheral blood, it could suggest a few possibilities such as immune activation, infection or pathological conditions, specific immune regulation. It is essential to note that the understanding of CD4+CD8+ T cells in the peripheral blood, especially in calves, is an area of ongoing research. Their role in peripheral blood may vary based on specific context and physiological conditions. We appreciate the reviewer's feedback in examining our work, and we are confident that our response will provide a more nuanced understanding of the observed variations in the CD4+CD8+ T cell subset representation.

4- is too strong as effector cell functions were not investigated.

Response: In response to your valuable feedback, I may not catch well the meaning of this comment. In this study we tried to evaluate the effector cell functions which are important for immune status of newborn JB calves in their first month of life. This evaluation could help farmers to select the best feeding method for newborn calves to reduce the morbidity and mortality rate during the first month of their lives.

5- what was the antibody and markers used for B lymphocytes?

Response: For B lymphocyte evaluation we used IgM and MHC class II antibodies, the information of antibodies was mentioned in (Table S2).

6- I noticed most of the antibodies are directly conjugated, but did the authors include isotype control

 Response: We did not use any isotype control in this study. However, in the past study, we evaluated the mouse IgG for isotype control which did not affect results of subsets analysis.

7-the caption for figure 1 appeared above the figure

Response: Thank you for your comment, we have fixed the caption for Figure 1.

8- how much formula replacer was given to each calf? How many times?

Response: The CR (Elanco Co., Tokyo, Japan) kept in refrigerator before using and was treated 750g three times (between 2 hr, between 6 hr and between 12 hr) after birth. This information was added in line 89-91.

9- Is it possible that these CR animals received a much more volume of colostrum replacer considering the capacity of milk production of cows?

Response: The volume of colostrum was adjusted according to amount of nutritional requirement for newborn Japanese black calves and all calves in each groups received same amount of colostrum without any exception.

10- could you also add the brand and origin of the tubes? Also, were the tubes of colostrum replacer kept refrigerated or at room temperature?

Response: The origin of tubes and their condition in farm was added in line 89-91.

11- Were all the cows multiparous?

Response: Yes, all the cows in this study were multiparous.

Reviewer 2 Report

Comments and Suggestions for Authors

Manuscript ID: animals-2688550:

Promotion of immunity, growth, and health of Japanese black calves by feeding maternal colostrum

The authors report a quite interesting study, which would have been straightforward if study design had been adequate. Unfortunately, study group and control group were kept on different farms, so that it is not possible to separate treatment effect from farm effect. Even though the results appear plausible, the study design does not allow the conclusions drawn.

Since the calves were from 4 farms there might be a possibility to include the farm effect into statistics, however, I’m not a statistician. 

Also, the English language is poor, especially in the introduction.

M&M section is insufficient with regards to the management of the calves, composition of MC and CR and further feeding (what does “the same” mean?)

Comments on the Quality of English Language

No further comments

Author Response

Reviewer2#

The authors report a quite interesting study, which would have been straightforward if study design had been adequate. Unfortunately, study group and control group were kept on different farms, so that it is not possible to separate treatment effect from farm effect. Even though the results appear plausible, the study design does not allow the conclusions drawn.

Since the calves were from 4 farms there might be a possibility to include the farm effect into statistics, however, I’m not a statistician. 

Also, the English language is poor, especially in the introduction.

M&M section is insufficient with regards to the management of the calves, composition of MC and CR and further feeding (what does “the same” mean?)

Response: Thank you for your comments, the four farms are not separated farms and they are related to one company in Miyazaki, Japan. All four farms are in single area and assigned to keep Japanese black calves from calving to market age. The company uses standard feeding system for Japanese black calves in Japan and due to the location of farms in a single area all four farms use the uniform management system which was mentioned in detail in line 95-109. Therefore, due to equal feeding and farm management system in all four farms, farm management would not affect treatment data and the effects tried to greatly be reduced.

For English proof reading, FORTE company helped us to edit this manuscript however by adding more information to manuscript some parts need to edit again. We did recheck the introduction part and edited it again and if still there is major problems with English editing, we can check it again by English proof-reading companies.

The composition of Colostrum and further feeding management was added in line 85-91 and 101-109 and (Table S1). The nutritional requirements are based on 8th edition of the Japanese beef cattle feeding standard (NRC, 2000).

Reviewer 3 Report

Comments and Suggestions for Authors

This research investigates the immune function, growth, and disease occurrence of colostrum fed to newborn calves: colostrum from mother cows and substitute colostrum. It contains a lot of useful information for managing the calves, and is expected to be used effectively in farms. However, there are some parts that are lacking in explanation, it is considered preferable to make additions and corrections and conduct the review again.

     L81: Were all 16 calves used in this research born naturally? It is known that dystocia are affected colostrum inoculation and absorption, and this may have affected the results. If there were no abnormalities at birth and no dystocia, it would be better to add a note to that effect.

     L82: Farms A and B are divided into MC group, and farms C and D are divided into CR group. The feeding plan is described in L85, but are the management methods (group or individual, etc.) also under the same conditions? Similarly, are there any differences in the feed management of mother cows before calving at each farm? It is thought that the nutritional management of the mother cow during the perinatal period influences the immune function of the neonatal period after calving. It may also affect the absorption of the colostrum fed, so if there is no difference, please state it.

     L82: It is known that colostrum absorption decreases in a time-dependent, but there is a difference in time between the MC group within 10 hours after birth and the CR group within 12 hours after birth. Does this time difference affect the results? Also, the usual colostrum feeding time is recommended to be within 6 hours, but why did it take longer than that? If there is no difference, please describe the time of colostrum inoculation for each group.

     L87: The history of respiratory disease in calves is compared only for one month. Blood samples are collected for up to 3 months, and if we are going to discuss the effects on immune function within 3 months, I think that the occurrence of disease should also be described up to 3 months.

     L89: Based on the carcass weight in 30 months of age at the slaughterhouse, growth promotion was observed in the MC group. Were they kept at farms A to D in L82 until they were 30 months old? Also, are there any differences between farms in the feeding management methods after the calf period? If the feeding management methods are different, is it possible that the difference in carcass weight is due to the difference in the feeding management method during the growing to fattening period, rather than the presence or absence of the colostrum feeding method?

     L173~: In lymphocyte subset analysis, each cell number is listed as a real number. I think the total white blood cell count is measured because the subsets are looked at as a percentage. Since the percentage expressed as a real number depends on the original measurement value (WBC), I think it is necessary to include WBC in the results.

     L276: It is described that the reason why there was no difference in phagocytosis between the two groups is thought to be because phagocytosis control factors were contained in both colostrum. However, the lifespan of neutrophils is generally shorter than that of lymphocytes, and is it possible that the influence of colostrum on the phagocytic effect has disappeared by the first month of life when the sample was collected?

Author Response

Reviewer3#

 This research investigates the immune function, growth, and disease occurrence of colostrum fed to newborn calves: colostrum from mother cows and substitute colostrum. It contains a lot of useful information for managing the calves, and is expected to be used effectively in farms. However, there are some parts that are lacking in explanation, it is considered preferable to make additions and corrections and conduct the review again.

①     L81: Were all 16 calves used in this research born naturally? It is known that dystocia is affected colostrum inoculation and absorption, and this may have affected the results. If there were no abnormalities at birth and no dystocia, it would be better to add a note to that effect.

Response: All registered calves in this study were born normally, and no abnormalities were noted by responsible staff and veterinarians during calving. We also mentioned it in line (91-92).

②     L82: Farms A and B are divided into MC group, and farms C and D are divided into CR group. The feeding plan is described in L85, but are the management methods (group or individual, etc.) also under the same conditions? Similarly, are there any differences in the feed management of mother cows before calving at each farm? It is thought that the nutritional management of the mother cow during the perinatal period influences the immune function of the neonatal period after calving. It may also affect the absorption of the colostrum fed, so if there is no difference, please state it.

Response: Thank you for your comments, the four farms are not separated farms and they are related to one company in Miyazaki, Japan and all four farms are in the same area. All four farms are assigned to keep Japanese black calves from calving to market age. One standard feeding system for Japanese black calves use for all mother cows and there are no differences in feeding management between same age cows in farms. Due to the location of farms in a single area the farmers use the same management system and same feeding standards which was mentioned in (Table S1) and line 95-109.

③     L82: It is known that colostrum absorption decreases in a time-dependent, but there is a difference in time between the MC group within 10 hours after birth and the CR group within 12 hours after birth. Does this time difference affect the results? Also, the usual colostrum feeding time is recommended to be within 6 hours, but why did it take longer than that? If there is no difference, please describe the time of colostrum inoculation for each group.

Response: The time of colostrum feeding was adjusted according to the amount of MC and CR and to make sure the calves get enough colostrum and successfully the passive immunity transferred by colostrum to newborn calves in each group we increased the time to 10 hr for MC and CR to 12 hr. However, this difference in time could not affect statistically the result and more information about the colostrum feeding was added in line 84-91.

④     L87: The history of respiratory disease in calves is compared only for one month. Blood samples are collected for up to 3 months, and if we are going to discuss the effects on immune function within 3 months, I think that the occurrence of disease should also be described up to 3 months.

Response: Japanese black calves in all four farms are registered calves and their diseases history record since they born. However, the first month of life in Japanese black calves are the critical time for their life. Most of the mortality and morbidity cases occur in the first month of life due to uncompleted passive transfer of immunity in newborn calves by colostrum. Also, one of the purposes of our study is to decrease the mortality and morbidity cases in 1 month old calves. Therefore, respiratory disease and other diseases which threaten the life of newborn calves were considered for the first month in this study.

⑤     L89: Based on the carcass weight in 30 months of age at the slaughterhouse, growth promotion was observed in the MC group. Were they kept at farms A to D in L82 until they were 30 months old? Also, are there any differences between farms in the feeding management methods after the calf period? If the feeding management methods are different, is it possible that the difference in carcass weight is due to the difference in the feeding management method during the growing to fattening period, rather than the presence or absence of the colostrum feeding method?

Response: The calves were housed for fattening in a separated fattening room till 30-month-old and the feeding amount for all cattle during fattening period was uniform. We added more information about feeding management and farm management in line 95-109.

⑥     L173~: In lymphocyte subset analysis, each cell number is listed as a real number. I think the total white blood cell count is measured because the subsets are looked at as a percentage. Since the percentage expressed as a real number depends on the original measurement value (WBC), I think it is necessary to include WBC in the results.

Response: We measured the WBC population for each calf, and we did add WBC graph in our result in (Figure S1) and line 204-206.

⑦     L276~: It is described that the reason why there was no difference in phagocytosis between the two groups is thought to be because phagocytosis control factors were contained in both colostrum. However, the lifespan of neutrophils is generally shorter than that of lymphocytes, and is it possible that the influence of colostrum on the phagocytic effect has disappeared by the first month of life when the sample was collected?

Response: Thank you for your thoughtful consideration. While it is plausible that the influence of colostrum on phagocytosis may diminish over time, it is important to note that the lasting effects on calf health can still be significant. Even if the observed phagocytic effects become comparable between the groups by the first month of life, the initial exposure to colostrum is crucial for establishing a robust immune foundation. Our data suggests no significant difference in phagocytosis between the MC and CR groups at the one-month mark. However, we believe that both groups still benefit from the early exposure to colostrum, contributing to overall calf health. Disappearing the phagocytosis effect causes severe diseases in one month old calves which leads to mortality and morbidity cases, especially when they are neonatal, and they get passive immunity by colostrum. Our hypothesis about phagocytosis result was based on previous research articles which measured phagocytosis on different time points by using different food additives in newborn animals.

Round 2

Reviewer 2 Report

Comments and Suggestions for Authors

Unfortunately, the authors did not address my concerns satisfactory

Comments on the Quality of English Language

Could be improved

Author Response

Thank you for investing your time and effort in reviewing our manuscript. We highly appreciate your insightful comments and valuable suggestions, which have significantly contributed to the improvement of our paper. In the revised version, all changes have been highlighted in red for your convenience. Below, you will find our detailed responses to each of the Reviewer's comments.

Reviewer2#

Unfortunately, the authors did not address my concerns satisfactory.

Quality of English Could be improved.

Response:

In response to the reviewers' concern, we acknowledge the potential influence of different farms on our study outcomes. It is important to note that all four farms (A, B, C, and D) involved in the study are geographically adjacent and adhere to uniform feeding and management practices. This intentional setup aims to minimize any farm-specific effects on our results. To address this concern explicitly, we have enhanced the Methods section by providing additional details on the standardized conditions across farms, including consistent feeding regimens, housing protocols, and healthcare practices. Furthermore, statistical analyses were employed to assess and control for potential farm-specific effects. These adjustments aim to offer readers a clearer understanding of how we managed farm variability and allow for a more robust interpretation of the treatment effects on immunity, health, and growth in JB calves. We appreciate the thorough review by the academic community, which has contributed to refining the clarity and rigor of our methodology.

For the material and methodology as you mentioned about:

  • Management of the calves, new section was added in line 83-97.
  • Composition of MC and CR, new section was added in line 100-111.
  • Further feeding information, a new section was added in line 114-123.
  • We have once again reviewed the manuscript for English language quality. We attached certificate of English editing service.

We hope the information provided adequately addresses your inquiry. Should you require further clarification, please do not hesitate to let us know. We highly value your feedback and are committed to ensuring a thorough understanding of our study through any additional information you may need."

Reviewer 3 Report

Comments and Suggestions for Authors

Many necessary explanations have been added to the revised draft, and the background of the calves in which the survey was conducted has been clarified. However, I did not explain my points well enough, as some of the content seems a bit difficult to accept, so I hope that they will reconsider.

     L109-112: As I pointed out before, blood samples are collected for up to 3 months, and if the effect on immune function is to be discussed within 3 months, the medical history should also be at least 3 months old. The response letter was written that, “one of the purposes of our study is to decrease the mortality and morbidity cases in 1 month old calves.” If so, I think the purpose should be stated in the introduction or discussion.

    L328-329: As I pointed out last time, the reason why there was no difference in phagocytosis between the two groups at 1 month of age is thought to be due to the effect that described in the response letter “the initial exposure to colostrum is crucial for establishing a robust immune foundation”. For this reason, I think it is necessary to state that in order to produce healthy calves with phagocytosis, it is necessary to ensure that colostrum is ingested.

Author Response

Thank you for investing your time and effort in reviewing our manuscript. We highly appreciate your insightful comments and valuable suggestions, which have significantly contributed to the improvement of our paper. In the revised version, all changes have been highlighted in red for your convenience. Below, you will find our detailed responses to each of the Reviewer's comments.

Reviewer3#

 L109-112: As I pointed out before, blood samples are collected for up to 3 months, and if the effect on immune function is to be discussed within 3 months, the medical history should also be at least 3 months old. The response letter was written that, “one of the purposes of our study is to decrease the mortality and morbidity cases in 1 month old calves.” If so, I think the purpose should be stated in the introduction or discussion.

Response:

Thank you for your subsequent feedback. In response to your comment, we have incorporated our research objective into the discussion, specifically in lines 387-390. In addition, we tried to correct the data about diseases at 3 months old calf from Miyazaki Agricultural Mutual Aid Association. We could not get the correct data because the management system of disease record changed.

  L328-329: As I pointed out last time, the reason why there was no difference in phagocytosis between the two groups at 1 month of age is thought to be due to the effect that described in the response letter “the initial exposure to colostrum is crucial for establishing a robust immune foundation”. For this reason, I think it is necessary to state that in order to produce healthy calves with phagocytosis, it is necessary to ensure that colostrum is ingested.

Response:

We express our gratitude for your valuable comment. After reviewing several articles, we have recognized that your point about the short lifespan of neutrophils serves as a strong support for our phagocytosis data. We have integrated the insightful feedback from your comment into the discussion section, specifically addressed in lines 342-345.